# The Diagnostic Accuracy of Magnetic Resonance Imaging Versus Transvaginal Ultrasound in Deep Infiltrating Endometriosis and Their Impact on Surgical Decision-Making: A Systematic Review

**DOI:** 10.3390/diagnostics15222856

**Published:** 2025-11-12

**Authors:** Michael O’Leary, Conor Neary, Edward Lawrence

**Affiliations:** Department of Radiology, University of Galway, H91 TK33 Galway, Ireland

**Keywords:** deep infiltrating endometriosis, transvaginal ultrasound, magnetic resonance imaging, diagnostic accuracy, surgical planning, multidisciplinary team, lesion mapping, women’s health

## Abstract

**Objectives:** This study aimed to systematically compare the diagnostic accuracy of magnetic resonance imaging (MRI) and transvaginal ultrasound (TVUS) for deep infiltrating endometriosis (DIE) and to evaluate their impact on surgical decision-making. **Methods:** We carried out a systematic review of studies (2015–2025) comparing MRI and TVUS for DIE in the same patients, with surgical/histological confirmation used as the reference standard. The primary outcomes were sensitivity and specificity by lesion site; the secondary outcomes included the reported impact on surgical decision-making and treatment planning. **Results:** Nine studies met the inclusion criteria. For rectosigmoid lesions, the sensitivity was 79–94% for TVUS and 86–94% for MRI, with a high specificity for both (84–95%). TVUS demonstrated strong diagnostic accuracy for posterior compartment disease, but its sensitivity was notably lower for uterosacral ligament and bladder lesions (25–83%). MRI showed higher sensitivity for anterior and multi-compartmental lesions (75–94%), reflecting its superior anatomical coverage. Imaging informed surgical planning, ensuring the inclusion of subspecialists such as colorectal or urological surgeons. It also guided the extent of resection and need for radical versus conservative procedures. **Conclusions:** TVUS and MRI are complementary imaging modalities in the diagnosis and staging of DIE. TVUS offers high specificity and remains particularly effective for posterior compartment disease, whereas MRI provides broader anatomical coverage and higher sensitivity for anterior and multi-compartmental involvement. Importantly, integrating imaging into preoperative planning improves multidisciplinary coordination, optimises preparedness, and guides resection. This is the first review to systematically compare MRI and TVUS for DIE with an emphasis on lesion-level accuracy and the impact on surgical decision-making.

## 1. Introduction

Endometriosis is a chronic gynaecological condition that affects up to 10% of women of reproductive age and can lead to pain, infertility, and reduced quality of life [1]. Deep infiltrating endometriosis (DIE) represents the most severe phenotype, characterised by lesions that infiltrate more than 5 mm beneath the peritoneum [2]. DIE arises through interacting endocrine, inflammatory, and immune-mediated mechanisms in which ectopic endometrial-like glands and stroma adhere to and invade pelvic structures, supported by local oestrogen production, progesterone resistance, and neuroangiogenesis. These processes lead to fibrosis, adhesion formation, and pain hypersensitivity, which explains the chronic and often progressive nature of the disease. Delayed diagnosis, which often spans many years, contributes to prolonged pain, reduced quality of life, and increased healthcare and productivity costs for affected women [1]. DIE frequently involves complex anatomical locations such as the uterosacral ligaments (USLs), rectosigmoid colon, bladder, and rectovaginal septum [3,4].

Accurate DIE diagnosis and staging are critical for determining the appropriate surgical approach. Surgery remains the gold-standard treatment for medically refractory or severe cases [5]. Preoperative imaging supports multidisciplinary surgical planning, identifying lesions that require colorectal or urological input, and ensuring informed consent for complex interventions [6]. In addition, transvaginal ultrasound and MRI assist in differentiating benign from malignant uterine lesions, thereby helping to exclude uterine sarcomas and guide appropriate surgical management [7].

MRI and TVUS are the primary imaging modalities for DIE evaluation. TVUS is the recommended first-line test for endometriosis due to its affordability and high accuracy when performed by an experienced practitioner [8]. However, MRI offers advantages in multiplanar imaging, superior soft-tissue contrast, and a large field of view, resulting in high diagnostic accuracy for detecting DIE [2]. Although the meta-analysis by Guerriero et al. (2018) primarily pooled diagnostic performance metrics from studies published up to 2017, it did not explore how imaging findings influence surgical planning, multidisciplinary coordination, or patient outcomes [3]. The present review provides an updated synthesis, incorporating evidence from 2017 onwards, analysing lesion-specific diagnostic accuracy across compartments, and examining how advances in imaging protocols and structured reporting affect operative decision-making.

The use of imaging remains inconsistent in the UK despite European guideline recommendations (ESHRE, 2022) [9]. UK NICE guidance [10] currently supports TVUS as the first-line imaging modality in suspected endometriosis and MRI in the setting of suspected deep disease, especially in preoperative assessment. Gaining an understanding of which lesion locations and specific patient groups might benefit from one or both imaging modalities could improve imaging utilisation and effectiveness.

## 2. Methods

The systematic review was conducted according to the Preferred Reporting Items for Systematic Reviews and Meta-Analyses (PRISMA) 2020 statement [11] and was prospectively registered with PROSPERO (CRD420251114636). The primary objective was to compare the diagnostic performance of TVUS and MRI for DIE using surgical and/or histopathological findings as the reference standard. The secondary objective was to evaluate the impact of both imaging modalities on surgical management, planning, and multidisciplinary team (MDT) coordination.

### 2.1. Search Strategy

We searched MEDLINE (through PubMed), EMBASE, Scopus, and the Cochrane Library from 1 January 2015 to 1 May 2025. We combined the following keywords and MeSH terms using Boolean operators: “deep infiltrating endometriosis”, “DIE”, “transvaginal ultrasound”, “TVUS”, “magnetic resonance imaging”, “MRI”, “diagnostic accuracy”, and “surgical planning”. The date range was chosen to capture contemporary studies reflecting modern imaging protocols and structured reporting standards. Earlier studies were excluded because they used outdated MRI or ultrasound techniques and predated the adoption of structured frameworks such as IDEA and ENZIAN. The complete set of Boolean search strings is available in Appendix A.

### 2.2. Eligibility Criteria

#### 2.2.1. Inclusion Criteria

Studies were eligible for inclusion if they involved women with suspected or confirmed deep infiltrating endometriosis (DIE) who underwent both transvaginal ultrasound (TVUS) and magnetic resonance imaging (MRI) within the same patient cohort. Laparoscopy (coelioscopy) with histological confirmation is considered the gold standard for diagnosing endometriosis [1]. Each study was required to use surgical and/or histopathological confirmation, including laparoscopy, as the reference standard and to report diagnostic accuracy outcomes, such as sensitivity and specificity, for both modalities. Studies were also included if they reported outcomes relevant to surgical decision-making, either directly (e.g., surgical planning or MDT referral) or indirectly (e.g., lesion location, depth, or lesion-to-anal-verge distance).

#### 2.2.2. Exclusion Criteria

Exclusion criteria comprised case reports, systematic or narrative reviews, meta-analyses, conference abstracts, or editorials. Studies were also excluded if they evaluated only a single imaging modality, lacked surgical or histopathological confirmation, or were not published in English.

### 2.3. Study Selection

Screening was conducted in two stages in Rayyan: titles and abstracts were screened, with full-text review of potentially eligible articles, and conflicts were resolved by consensus. We report the selection process using a PRISMA flowchart.

### 2.4. Data Extraction

A standardised form was used to extract data: author, year, study design, sample size, imaging protocols, anatomical compartments evaluated, reference standards, and reported diagnostic accuracy metrics (sensitivity and specificity). For imaging protocols, extracted details included MRI field strength, sequence type, bowel preparation, the use of vaginal or rectal gel, and the application of structured frameworks such as the International Deep Endometriosis Analysis (IDEA) or ENZIAN classification, where reported. The reported impact on surgical decision-making was also extracted if applicable (e.g., change in planned procedure, MDT referral). Lesion sites were categorised according to the anatomical compartments most commonly reported across studies, which correspond to the organ-based regions defined within the IDEA consensus framework. When studies reported pooled diagnostic outcomes without compartmental breakdown, these were included in the overall synthesis but excluded from compartment-specific subgroup comparisons to preserve analytical consistency.

### 2.5. Quality Assessment

Study quality and risk of bias were evaluated using the QUADAS-2 tool [12]. It assesses four domains: patient selection, index test, reference standard, and flow/timing. Two reviewers independently evaluated each study, and disagreements were resolved by consensus or, if required, by a third reviewer. Overall observational quality was cross-checked using the Newcastle-Ottawa Scale. Both of these assessments are available in Appendix A. Bias between treatment and non-treatment groups was not applicable, as all included studies evaluated diagnostic accuracy rather than therapeutic or interventional outcomes.

### 2.6. Synthesis of Results

Due to the substantial heterogeneity in imaging protocols, study design, reference standards, and compartment definitions, a formal meta-analysis or pooled quantitative synthesis was not feasible. Instead, the results were organised by anatomical compartment (posterior, anterior, bladder, rectovaginal, uterosacral) to allow for a descriptive comparison of MRI and TVUS performance. Where possible, averaged sensitivity and specificity values are summarised visually in radar plots to enhance transparency and interpretability. The criteria for inclusion in the synthesis were based on study design, risk of bias assessment, and relevance to the review question. Subgroup analyses highlighted variability in sensitivity and specificity according to imaging technique, protocol optimisation, and operator expertise. In addition, reported data on the impact of imaging on surgical decision-making were extracted where available to provide insights into the role of preoperative imaging in multidisciplinary planning.

## 3. Results

### 3.1. Literature Search and Study Selection

An initial search resulted in 242 unique records. All citations were exported to Rayyan, and duplicates were removed. After screening, nine studies were identified that satisfied the inclusion standards and covered prospective and retrospective comparative diagnostic accuracy study formats. A flow diagram of the study selection process is provided in Figure 1.

### 3.2. Study Characteristics

All studies reported on TVUS and MRI performed on the same cohort and had surgical and/or histopathological confirmation as the reference standard. Study characteristics are summarised in Table 1. Sample sizes across the nine included studies ranged from approximately 40 to 178 participants (median ≈ 90), reflecting differences in referral patterns and centre volume. This variation in study size likely contributed to the wide range of reported sensitivities and specificities across compartments and modalities, as smaller series tended to show broader confidence intervals and greater variability.

### 3.3. Technical Protocol Variability

As detailed in Appendix A, imaging protocols varied considerably among the included studies. Only three studies explicitly applied elements of the International Deep Endometriosis Analysis (IDEA) or ENZIAN frameworks. Bowel preparation ranged from none to full enema or rectal contrast administration. MRI protocols differed in field strength (1.5 T vs. 3 T), slice thickness (3–4 mm vs. variable), and sequence composition, with some incorporating fat-suppressed T2-weighted imaging, vaginal and rectal gel, or intravenous gadolinium contrast. These variations in technique and operator expertise likely contributed to the heterogeneity in diagnostic performance observed across anatomical compartments.

### 3.4. Quality Assessment

The overall methodological quality was moderate to high. Risk of bias was low for key domains, including reference standard and flow and timing. Variability was noted in the description of patient selection methods and the blinding of index tests, which is to be expected in studies conducted in real-world settings with different study designs. Applicability concerns were considered low in the majority of studies, as most were performed by utilising recognised diagnostic pathways.

All included studies achieved moderate-to-high methodological quality according to the Newcastle–Ottawa Scale, with scores ranging from 6 to 8 out of a maximum of 9. Most studies were rated highly for selection and outcome domains, reflecting representative patient inclusion and clear diagnostic endpoints. Minor limitations were observed in the comparability domain, largely due to variability in the reporting of operator expertise and imaging protocols.

### 3.5. Diagnostic Accuracy

The diagnostic accuracy of MRI and TVUS across anatomical compartments is summarised in Table 2. Comparisons were descriptive rather than inferential, as studies varied widely in their design, population, and reporting format, and few provided uniform numerators or confidence intervals required for formal statistical testing. A graphical comparison of MRI and TVUS diagnostic performance is presented in Appendix A.

Of the nine studies, seven reported conventional diagnostic accuracy metrics (sensitivity and specificity) and are included in Table 2. Two studies evaluated non-traditional diagnostic endpoints and were therefore excluded from the table: one assessing lesion-to-anal-verge distance (LAVD) mapping accuracy, and another evaluating correct multidisciplinary team assignment. These are discussed separately in Section 3.6 due to their distinct methodological focus.

Sample sizes varied widely across studies, ranging from 40 to 178 participants. Smaller cohorts tended to report greater variability in sensitivity and specificity, likely reflecting sampling bias and reduced statistical power, whereas larger, single-centre studies showed more stable accuracy estimates.

Both TVUS and MRI offer comparably high diagnostic accuracy for rectosigmoid involvement, with reported sensitivities of 79–94% for TVUS and 86–94% for MRI, and specificities of 84–95% for both modalities [15,16,19]. One study reported a notable drop in TVUS specificity to 50% [20], likely due to a high rate of false positives for posterior compartment lesions.

MRI consistently demonstrated superior diagnostic accuracy compared to TVUS for uterosacral ligament disease across most comparative studies. The sensitivity of MRI for USL lesions was consistently reported as 75–94%, whereas the sensitivity of TVUS was variable between 25 and 83%. Specificity was high for both modalities. The lowest sensitivity for USL lesions (25%) was observed in a study using standard 2D imaging [18] without bowel preparation, gel contrast, or a structured protocol. In contrast, higher sensitivities were reported in studies where experienced operators used standardised approaches, namely the IDEA protocol in one study [16] and 3D TVUS in another study [15]. These findings suggest that MRI is generally more reliable for parametrial and deep posterior lesions, particularly when TVUS is performed without advanced techniques.

Anterior compartment (bladder) lesions were less frequently reported, although MRI was generally preferred. In the two studies that reported anterior site prevalence, MRI sensitivity ranged from 50% to 100%, whereas TVUS sensitivity ranged from 25% to 89% [15,16]. The lower TVUS sensitivity found in one study [15] likely reflects the modality’s limited anterior visualisation, while higher values in another were associated with protocol optimisation and experienced operators [16].

For rectovaginal septum disease, both modalities showed good agreement, with MRI performing slightly better. The reported sensitivities for MRI ranged from 83% to 88% compared with 67% to 73% for TVUS, with similar specificities (93–100%) [15,16].

Two studies without compartment-specific breakdowns reported overall high DIE diagnostic accuracy, with one showing higher sensitivity and specificity for MRI (91% and 85%) compared with TVUS (77% and 70%) [21], and another finding similar performance between 3D rectosonography and MRI for rectosigmoid lesions, though MRI had a lower specificity [17].

Across studies reporting confidence intervals, most demonstrated narrow 95% CIs (<10% width), indicating high precision and stable diagnostic performance estimates. Wider intervals in smaller or single-centre studies [21] suggest greater uncertainty due to limited sample sizes or protocol variability. Full confidence interval data for each study are provided in Appendix A.

### 3.6. Impact on Surgical Planning

Several studies investigated the role of imaging in preoperative decision-making and surgical logistics. One study corroborated the usefulness of integrating multiple imaging modalities, showing that concordance between two or more diagnostic methods (clinical exam, TVUS, and MRI) significantly increased diagnostic confidence [19].

MRI and TVUS were both effective in detecting DIE with full-thickness rectal wall infiltration, an important finding for surgical planning as this typically necessitates segmental resection rather than conservative surgery [20]. MRI achieved both a higher specificity and positive predictive value (100%), while TVUS demonstrated a slightly higher sensitivity (93.6% vs. 86.4%) but a lower specificity (50%) [20]. This suggests a role for TVUS in initial detection and for MRI in confirmation when planning high-risk surgery. Neither modality could reliably differentiate between patients suitable for conservative bowel surgery, such as shaving or discoid excision, suggesting a possible limit in their ability to fully guide the surgical approach in cases not requiring segmental resection.

Another study demonstrated high sensitivity for both MRI (87%) and 3D rectosonography (93%) in assessing rectosigmoid infiltration [17]. The study did not specifically correlate imaging findings with type of resection; however, given the importance of assessing the depth and extent of lesions in planning the optimal surgical approach, these data support the use of preoperative imaging in the context of surgical planning to distinguish more conservative from radical surgical procedures.

Lesion-to-anal-verge distance mapping, vital for colorectal risk stratification and anastomotic planning, was evaluated in a further study [13]. Accurate estimation of lesion-to-anal-verge distance mapping informs the feasibility of safe primary anastomosis, the risk of temporary stoma formation, and preoperative counselling. TVUS and MRI using central calliper measurements both achieved ~70% accuracy within ±20 mm of intraoperative measurements, while MRI direct-calliper readings slightly underestimated distance without significantly affecting surgical planning.

One study applied the International Deep Endometriosis Analysis (IDEA) consensus framework across TVUS, MRI, and intraoperative reporting, establishing a unified terminology for lesion localisation and depth assessment [16]. This standardisation enabled consistent anatomical referencing between modalities and yielded high inter-modality agreement (κ = 0.727 for TVUS; κ = 0.755 for MRI). Although the sample size limited formal validation, these findings support the feasibility of structured reporting to enhance interdisciplinary communication and surgical planning.

In another study, the addition of MRI to the standard work-up improved correct multidisciplinary team assignment from 71.6% to 90.5% (+18.9 percentage points), further supporting the value of imaging in accurately mapping disease distribution, predicting operative complexity, optimising consent, and ensuring appropriate subspecialist support in theatre [14].

## 4. Discussion

Nine studies were included in this systematic review, all comparing TVUS and MRI in the same patient cohort for DIE. The overall diagnostic performance of both modalities was good to excellent, with variability based on lesion location. The findings support a complementary role for the two modalities, with TVUS as the first-line option for direct, wide-access examination, given accessibility and cost, and MRI as a useful adjunct tool for comprehensive or targeted mapping. Several studies pointed out the role of imaging beyond diagnosis, such as preoperative imaging’s effect on changing surgical strategy, improving multidisciplinary triage, and allowing for more thorough patient counselling.

### 4.1. Comparative Diagnostic Accuracy

MRI and TVUS both fared well in this analysis for rectosigmoid DIE, with comparable accuracy in many cases, but a modest sensitivity advantage for MRI in settings where adjunctive techniques for TVUS were not available (e.g., rectal gel, 3D reconstructions, or both). Similarly, the specificity of TVUS showed wider variability, with one study reporting an increased false positive rate in a high-prevalence cohort, likely due to the difficulty in distinguishing posterior compartment lesions from adjacent adhesions or non-invasive fibrosis in the absence of the optimal technique [20]. This analysis therefore supports prior conclusions that TVUS in expert hands is a highly sensitive and specific modality. The principal limitations of TVUS stem from variability in operator skill and adherence to technique, rather than from the intrinsic capabilities of the modality itself. MRI’s advantage reflects its reproducibility and capacity to provide a comprehensive anatomical overview, rather than its intrinsically higher sensitivity. These findings also support a complementary strategy of multi-modality imaging to provide greater confidence in reaching a diagnosis or for more extensive or multi-compartmental disease.

MRI was more accurate than TVUS in USL and bladder involvement, where TVUS performance was more heterogeneous. In one study, the sensitivity of TVUS for USL disease was particularly low at 25% [18], in contrast with sensitivities of 73–83% observed in other studies included in this review [15,16,19]. This poorer performance is likely the result of a lack of a standardised protocol, non-utilisation of adjunctive techniques (e.g., gel sonovaginography), and the inherent difficulty of parametrial mapping outside of a specialist or high-volume practice.

### 4.2. Multidisciplinary Planning and Surgical Logistics

A number of studies highlighted the importance of imaging in surgical logistics and multidisciplinary planning. In one study, multidisciplinary team stratification accuracy improved by nearly 19 percentage points when MRI was added to TVUS, ensuring that appropriate surgical personnel were present based on imaging findings [14].

Another study demonstrated that combining more than one imaging modality, or adding clinical examination, enhanced diagnostic accuracy and the mapping of disease distribution, particularly in the posterior and central compartments, where complete resection and team coordination are critical [19].

MRI and TVUS were also reported to have similar accuracy in estimating lesion-to-anal-verge distance (LAVD), an important parameter for preoperative colorectal risk assessment [13]. Accurate LAVD estimation supports anastomotic planning and may reduce intraoperative uncertainty.

In addition, MRI was consistently valuable for defining lesion extent and informing decisions regarding bowel and bladder surgery [17,20]. By enabling accurate preoperative mapping and correct multidisciplinary team allocation, advanced imaging reduces intraoperative uncertainty and the likelihood of unplanned conversions or incomplete resections, ultimately supporting safer surgery and improved postoperative outcomes. Taken together, these findings suggest that complementary use of MRI and TVUS can support surgical decision-making and improve theatre preparedness.

A schematic summary of the proposed diagnostic pathway is presented in Figure 2.

### 4.3. Protocol Variability and the Need for Standardisation

Although strong diagnostic performance was demonstrated, heterogeneity in imaging protocols was the most consistent limitation across the included studies. Only a minority utilised standardised frameworks, such as the International Deep Endometriosis Analysis (IDEA) protocol for TVUS, and MRI protocols varied in the use of bowel preparation, gel contrast, and fat suppression. These differences likely influenced diagnostic performance, with anterior compartment sensitivity varying between studies [15,16]. Lower sensitivity was reported in a multicentre series despite the use of 2D/3D TVUS [15], whereas higher sensitivity was seen when the IDEA framework and optimised anterior protocols were applied [16], highlighting the impact of protocol optimisation and operator experience rather than the use of 3D imaging alone.

This variability is also evident in UK practice. A national audit of British Society of Gynaecological Endoscopy centres reported wide heterogeneity in MRI staging protocols, with inconsistent use of fasting, antispasmodics, sequence acquisition, and limited adoption of rectal/vaginal opacification [22]. This variability also reflects limited adoption of the European Society of Urogenital Radiology (ESUR) standards which recommend multiplanar high-resolution T2-weighted imaging, appropriate bowel preparation, bladder filling, and the use of antiperistaltic agents [23]. Such inconsistency poses a barrier to reproducibility and supports the need for national protocol standardisation, particularly as imaging increasingly replaces diagnostic laparoscopy in NHS pathways.

### 4.4. Structured Reporting and Innovation

Recent developments have introduced practical solutions to help overcome variability and improve reproducibility. Abbreviated MRI protocols (aMRI) have been evaluated, showing comparable diagnostic accuracy to full protocols while reducing acquisition time by approximately 30% and costs by nearly 45% [24]. This makes aMRI an attractive option for healthcare systems such as the NHS, where efficiency is critical. In parallel, structured MRI-based classification systems for uterosacral ligament (USL) lesions have been proposed [25], offering a reproducible framework that may improve diagnostic confidence and inter-observer agreement. Together, these innovations represent feasible strategies to enhance consistency in reporting and facilitate wider clinical integration of imaging in endometriosis care. Comparable initiatives in the U.S., such as the American College of Radiology (ACR) Appropriateness Criteria for endometriosis, which endorse structured MRI reporting shown to improve sensitivity and clinician preference, highlight a parallel trend toward standardisation [26].

Additionally, recent work using deep learning on multi-sequence MRI has shown that AI-assisted models can modestly improve radiologist sensitivity and inter-observer agreement in detecting endometriosis, highlighting their potential to enhance diagnostic consistency and efficiency in future practice [27].

### 4.5. Future Directions and Relevance

Future work should prioritise multicentre studies to assess the real-world impact of imaging standardisation and structured reporting on surgical outcomes and patient experience. In parallel, more structured approaches could help reduce diagnostic variation and support more efficient use of NHS imaging resources. Although NICE Guidelines have broadly endorsed imaging for suspected DIE, there is still limited detailed protocol guidance. Future iterations could draw on the recent radiology literature [22,24,25], which has examined protocol standardisation, abbreviated MRI approaches, and lesion classification systems, to promote more consistent clinical practice and service delivery.

### 4.6. Limitations

This review has several limitations. There was significant heterogeneity among the included studies regarding imaging protocols, operator experience, and reporting criteria, which may have influenced diagnostic performance and limited comparability. Blinding procedures were inconsistently described, introducing a potential risk of interpretation bias. Most studies were performed in expert tertiary centres with specialised imaging personnel, which may overestimate diagnostic accuracy compared with general practice. Finally, publication bias cannot be excluded, as studies reporting favourable diagnostic performance are more likely to be published.

## 5. Conclusions

Both MRI and transvaginal ultrasound demonstrate high diagnostic accuracy for deep infiltrating endometriosis. TVUS remains the first-line imaging modality, particularly effective for posterior compartment disease when performed by an experienced operator. MRI offers complementary value through superior reproducibility and comprehensive evaluation of anterior and multi-compartment lesions, guiding multidisciplinary surgical planning. Greater protocol standardisation and the integration of structured reporting may further improve diagnostic consistency and clinical utility.

## Figures and Tables

**Figure 1 diagnostics-15-02856-f001:**
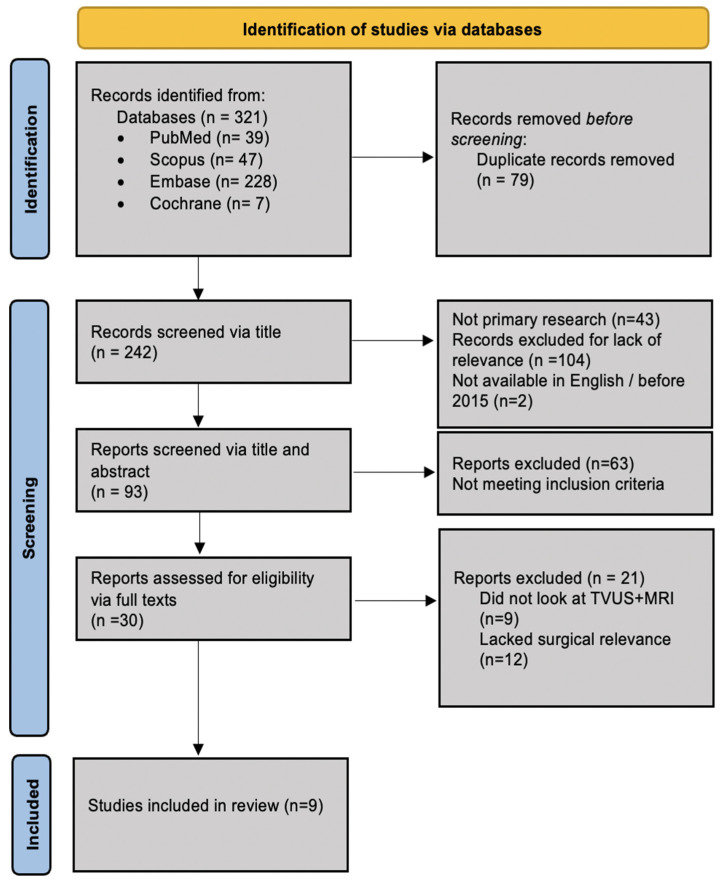
PRISMA flowchart.

**Figure 2 diagnostics-15-02856-f002:**
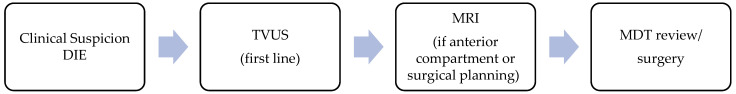
The suggested imaging and multidisciplinary pathway for suspected deep infiltrating endometriosis (DIE). TVUS is performed as the first-line investigation. MRI is recommended for anterior or multi-compartment disease, or when detailed preoperative mapping is required. Imaging findings guide multidisciplinary team (MDT) discussion and surgical planning.

**Table 1 diagnostics-15-02856-t001:** Summary of included studies.

Author (Year)	Study Location	Study Size (*n*)	Study Type	Single- vs. Multicentre	Reference Standard
Aas-Eng et al. (2023) [13]	Norway	47	Prospective observational study	Single	Intraoperative measurement (rectal probe)
Bielen et al. (2020) [14]	Belgium	74	Prospective observational study	Single	Laparoscopy findings (intraoperative)
Guerriero et al. (2017) [15]	Italy and Spain	159	Prospective observational study	Multi	Surgical histology
Indrielle-Kelly et al. (2020) [16]	UK	49	Prospective observational cohort	Multi	Intraoperative mapping + histopathology
Philip et al. (2020) [17]	France	101	Prospective cohort study	Single	Surgery ± pathology
Puri et al. (2022) [18]	India	40	Prospective study	Single	Histopathology (laparoscopy)
Roditis et al. (2023) [19]	France	178	Retrospective diagnostic accuracy study	Multi	Surgery + histology
Sloss et al. (2022) [20]	Australia	52	Retrospective cohort	Single	Histology of resection specimens
Zaidi et al. (2023) [21]	Pakistan	90	Prospective comparative study	Multi	Laparoscopy + histopathology

**Table 2 diagnostics-15-02856-t002:** Diagnostic accuracy (sensitivity, specificity) based on location and modality. For clarity and to avoid blank fields, Table 2 has been divided into five sub-tables (**a**–**d**), each presenting diagnostic accuracy results for a specific anatomical compartment, and (**e**) showing the overall diagnostic accuracy from the study that did not report compartment-specific data.

(**a**) **Rectosigmoid Lesions**				
**Study (Year)**	**TVUS Sensitivity (%)**	**TVUS Specificity (%)**	**MRI Sensitivity (%)**	**MRI Specificity (%)**
Indrielle-Kelly [16]	94	84	94	84
Roditis [19]	79	95	86	90
Guerriero [15]	85	87	92	95
Sloss [20]	94	50	86	100
Philip [17]	93	95	87	90
(**b**) **Uterosacral Ligament Lesions**				
**Study (Year)**	**TVUS Sensitivity (%)**	**TVUS Specificity (%)**	**MRI Sensitivity (%)**	**MRI Specificity (%)**
Indrielle-Kelly [16]	74	67	94	60
Roditis [19]	83	92	83	83
Guerriero [15]	73	87	88	83
Puri [18]	25	97	75	100
(**c**) **Bladder Lesions**				
**Study (Year)**	**TVUS Sensitivity (%)**	**TVUS Specificity (%)**	**MRI Sensitivity (%)**	**MRI Specificity (%)**
Indrielle-Kelly [16]	89	100	100	95
Guerriero [15]	25	98	50	97
(**d**) **Rectovaginal Septum Lesions**				
**Study (Year)**	**TVUS Sensitivity (%)**	**TVUS Specificity (%)**	**MRI Sensitivity (%)**	**MRI Specificity (%)**
Indrielle-Kelly [16]	67	100	83	93
Guerriero [15]	73	87	88	83
(**e**) **Non-Compartment-Specific**				
**Study (Year)**	**TVUS Sensitivity (%)**	**TVUS Specificity (%)**	**MRI Sensitivity (%)**	**MRI Specificity (%)**
Zaidi [21]	77	70	91	85

## Data Availability

All data are derived from publicly available published studies.

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
