# Peer review of "The Diagnostic Accuracy of Magnetic Resonance Imaging Versus Transvaginal Ultrasound in Deep Infiltrating Endometriosis and Their Impact on Surgical Decision-Making: A Systematic Review"

_diagnostics, 2025, doi:10.3390/diagnostics15222856_

Round 1
Reviewer 1 Report
Comments and Suggestions for Authors
Dear Authors,
Thank you very much for allowing me to express my opinions related to your work. As a researcher myself, I admire and respect the effort you put into constructing your study and building this manuscript.
Bellow, you can find my comments regarding certain issues. I hope these comments will help you improve both your current and future work.
Introduction
- The background is generally adequate and shows well the clinical importance of accurate DIE imaging, although a bit more depth on the pathophysiological background could help readers outside gynecologic imaging.
- The reasoning for comparing MRI and TVUS is solid, but the novelty could be expressed more clearly compared with previous meta-analysis like Guerriero et al., UOG 2018.
- The introduction might also benefit from a mention about the socioeconomic consequences of delayed diagnosis, and why preoperative mapping improves not only surgical but also multidisciplinary coordination.
Methods
- The study design follows PRISMA guidelines and overall uses proper inclusion criteria, however some more technical details are missing, especially regarding MRI and TVUS protocols (slice thickness, bowel prep, use of gel contrast, sequence selection etc.).
- The search strategy is described, but should ideally show the full Boolean structure and list all databases — maybe this could be added as a supplementary table.
- It is not fully clear if inter-observer agreement was assessed during QUADAS-2 evaluation or how discrepancies were solved, which slightly limits reproducibility.
- The authors justified not doing a meta-analysis due to heterogeneity, which is fair, but even a simple pooled or compartment-based quantitative synthesis would make the results stronger and more transparent.
- The description of lesion compartment classification could also be more explicit, clarifying if the posterior, anterior, and rectovaginal sites were analyzed separately or mixed in some datasets.
Results
- The results are well organized and readable, but Table 2 would benefit from adding 95% confidence intervals for sensitivity and specificity; this omission makes the interpretation less robust.
- Sample sizes vary widely between studies (from 40 up to 178 cases), and this heterogeneity should be commented in terms of diagnostic weighting or effect on accuracy.
- A graphical illustration such as a forest plot or a radar-style chart comparing MRI versus TVUS by compartment would make the findings easier to visualise.
- The section discussing the impact of imaging on surgical planning is probably the strongest point of the paper; still, it could be expanded by adding quantitative data if available, like the proportion of cases where surgical plan or MDT composition changed after imaging.
Discussion
- The discussion is complete and demonstrates good knowledge of the field, but in places it becomes slightly repetitive and culd be shortened.
- The complementary nature of MRI and TVUS is explained well, but the paragraph should distinguish clearer between operator-dependent versus technique-related limitations of TVUS.
- MRI’s advantage comes more from reproducibility and global anatomical overview rather than intrinsic sensitivity; this nuance could be made more visible.
- The section about protocol variability is excellent and clinically relevant. It would add value if the authors include a few practical tips, for example minimal MRI protocol standards, the role of bowel preparation, or which T2 planes are most informative.
- The section on structured reporting and innovation (IDEA, Hôtel-Dieu classification, abbreviated MRI) is very good and aligns with modern trends, though a brief note on AI-assisted or radiomics-based analysis would make it more forward-looking.
- The limitations part should more clearly acknowledge that most studies were performed in tertiary centers and that publication bias may favor high-performing sites.
Conclusion
- The take-home message could be phrased in a more concise and practical way, emphasizing that TVUS remains the first-line approach for posterior compartment disease, while MRI provides superior assessment for anterior and multi-compartment involvement.
- Including a brief schematic summary or a simple decision flow (e.g. TVUS → MRI → MDT) would increase the practical usability of the paper.
Thank you very much for allowing me to express my opinions.
Sincerely.
Author Response
6/11/2025
RE: diagnostics-3967811
TITLE: Diagnostic Accuracy of Magnetic Resonance Imaging Versus Transvaginal Ultrasound in Deep Infiltrating Endometriosis and Their Impact on Surgical Decision-Making: A Systematic Review
Dear Reviewer,
We would like to sincerely thank you for your thoughtful and constructive feedback on our manuscript. The comments were highly insightful and have significantly improved the quality and clarity of our work.
In response, we have carefully revised the manuscript and prepared a detailed, point-by-point reply outlining how each comment has been addressed. All changes made in response to the reviewers’ comments are shown in red text, while any tracked changes represent English language refinements performed by the MDPI Author Services editing team.
We are very grateful for the time and expertise you and the editorial team devoted to reviewing our manuscript, and we hope that the revised version now meets the expectations of Diagnostics.
Thank you again for your consideration. Please do not hesitate to contact us if any further information is required.
Respectfully,
Michael O’Leary, Conor Neary, Edward M Lawrence, MD/PhD
Reviewer 1:
Comments:
Introduction
- The background is generally adequate and shows well the clinical importance of accurate DIE imaging, although a bit more depth on the pathophysiological background could help readers outside gynecologic imaging.
Reply: Thank you for pointing this out. We agree with this comment. Therefore, we have decided to include a concise description of the endocrine, inflammatory, and immune-mediated mechanisms underlying deep infiltrating endometriosis. This can be found on page 2, section 1 line 42-47.
“DIE arises through interacting endocrine, inflammatory, and immune-mediated mechanisms in which ectopic endometrial-like glands and stroma adhere to and invade pelvic structures, supported by local estrogen production, progesterone resistance, and neuroangiogenesis. These processes lead to fibrosis, adhesion formation, and pain hypersensitivity that explain the chronic and often progressive nature of the disease.”.
- The reasoning for comparing MRI and TVUS is solid, but the novelty could be expressed more clearly compared with previous meta-analysis like Guerriero et al., UOG 2018.
Reply: We revised the third paragraph of the Introduction page 3, lines 63–69 to clarify how this review extends prior work.
“Whereas the meta-analysis by Guerriero et al. (2018) primarily pooled diagnostic performance metrics from studies published up to 2017, it did not explore how imaging findings influence surgical planning, multidisciplinary coordination, or patient outcomes. The present review provides an updated synthesis, incorporating evidence from 2017 onwards, analysing lesion-specific diagnostic accuracy across compartments, and examining how advances in imaging protocols and structured reporting affect operative decision-making”
- The introduction might also benefit from a mention about the socioeconomic consequences of delayed diagnosis, and why preoperative mapping improves not only surgical but also multidisciplinary coordination.
Reply: A statement on diagnostic delay and its socioeconomic impact has been added to the Introduction on page 2, lines 47-49.
“Delayed diagnosis, which often spans many years, contributes to prolonged pain, reduced quality of life, and increased health-care and productivity costs for affected women”
Methods
- The study design follows PRISMA guidelines and overall uses proper inclusion criteria, however some more technical details are missing, especially regarding MRI and TVUS protocols (slice thickness, bowel prep, use of gel contrast, sequence selection etc.).
Reply: In the Results Section we have added a section on page 6, lines 170-179 on technical protocol variability and provided a more in depth table in Supplementary Table S3.
“As detailed in Supplementary Table S3, imaging protocols varied considerably among the included studies. Only three studies explicitly applied elements of the International Deep Endometriosis Analysis (IDEA) or ENZIAN frameworks. Bowel preparation ranged from none to full enema or rectal contrast administration. MRI protocols differed in field strength (1.5 T vs 3 T), slice thickness (3–4 mm vs variable), and sequence composition, with some incorporating fat-suppressed T2-weighted imaging, vaginal and rectal gel, or intravenous gadolinium contrast. These variations in technique and operator expertise likely contributed to the heterogeneity in diagnostic performance observed across anatomical compartments.”
- The search strategy is described, but should ideally show the full Boolean structure and list all databases — maybe this could be added as a supplementary table.
Reply: Thank you, we have added the full Boolean Search Strings to Supplementary Table S2.
- It is not fully clear if inter-observer agreement was assessed during QUADAS-2 evaluation or how discrepancies were solved, which slightly limits reproducibility.
Reply: We have now addressed this on page 4 lines, 133-136.
“Disagreements were resolved by consensus or, if required, by a third reviewer. Overall observational quality was cross-checked using the Newcastle–Ottawa Scale. Both these assessments are available in supplementary tables.”
- The authors justified not doing a meta-analysis due to heterogeneity, which is fair, but even a simple pooled or compartment-based quantitative synthesis would make the results stronger and more transparent.
Reply: We agree with the reviewer that a compartment-based quantitative summary improves clarity. While statistical pooling was not feasible owing to heterogeneity, we revised Section 2.6 page 4 lines 140-144 to state this explicitly and included radar plots to illustrate compartment-level performance.
“Due to substantial heterogeneity in imaging protocols, study design, reference standards, and compartment definitions, a formal meta-analysis or pooled quantitative synthesis was not feasible. Instead, results were organised by anatomical compartment (posterior, anterior, bladder, rectovaginal, uterosacral) to allow descriptive comparison of MRI and TVUS performance”
- The description of lesion compartment classification could also be more explicit, clarifying if the posterior, anterior, and rectovaginal sites were analyzed separately or mixed in some datasets.
Reply: We thank the reviewer for this important observation. We have clarified in the Methods → Data Extraction Section that studies reporting pooled diagnostic outcomes without compartmental separation were included in the overall synthesis but excluded from compartment-specific comparisons. This ensures transparency regarding dataset heterogeneity and analytical consistency. This clarification can be found in Section 2.4 Data Extraction on page 3, lines 126-129.
“ When studies reported pooled diagnostic outcomes without compartmental breakdown, these were included in the overall synthesis but excluded from compartment-specific subgroup comparisons to preserve analytical consistency.”
Results
- The results are well organized and readable, but Table 2 would benefit from adding 95% confidence intervals for sensitivity and specificity; this omission makes the interpretation less robust.
Reply: We have now included a section on page 8+9, lines 241-245, discussing confidence intervals where applicable.
“Across studies reporting confidence intervals, most demonstrated narrow 95 % CIs (<10 % width), indicating high precision and stable diagnostic performance estimates. Wider intervals in smaller or single-centre studies (e.g., Zaidi et al., 2023) suggest greater uncertainty due to limited sample sizes or protocol variability. Full confidence interval data for each study are provided in Supplementary Table S7.”
We have also included an expanded confidence interval table in Supplementary Tables S7.
- Sample sizes vary widely between studies (from 40 up to 178 cases), and this heterogeneity should be commented in terms of diagnostic weighting or effect on accuracy.
Reply: We appreciate the reviewer’s observation. We have expanded Section 3.5: Diagnostic Accuracy, page 8 lines 208-211 to comment on how variation in study sample size may have influenced diagnostic weighting and reported accuracy. Specifically, we note that smaller studies demonstrated greater variability in sensitivity and specificity, likely due to sampling bias and limited statistical power.
“Sample sizes varied widely across studies, from 40 to 178 participants. Smaller cohorts tended to report greater variability in sensitivity and specificity, likely reflecting sampling bias and reduced statistical power, whereas larger, single-centre studies showed more stable accuracy estimates.”
- A graphical illustration such as a forest plot or a radar-style chart comparing MRI versus TVUS by compartment would make the findings easier to visualise.
Reply: We have addressed this on page 7, lines 197-199 and have included radar charts reflecting the results in the Supplementary Materials S6.
- The section discussing the impact of imaging on surgical planning is probably the strongest point of the paper; still, it could be expanded by adding quantitative data if available, like the proportion of cases where surgical plan or MDT composition changed after imaging.
Reply: We expanded this section on page 9, lines 273-284 to include further quantitative data.
“One study applied the International Deep Endometriosis Analysis (IDEA) consensus framework across TVUS, MRI, and intraoperative reporting, establishing a unified terminology for lesion localisation and depth assessment [16]. This standardisation enabled consistent anatomical referencing between modalities and yielded high inter-modality agreement (κ = 0.727 for TVUS; κ = 0.755 for MRI). Although the sample size limited formal validation, these findings support the feasibility of structured reporting to enhance interdisciplinary communication and surgical planning.
In one study, addition of MRI to the standard work-up improved correct multidisciplinary-team assignment from 71.6 % to 90.5 % (+18.9 percentage points), further supporting the value of imaging in accurately mapping disease distribution, predicting operative complexity, optimising consent, and ensuring appropriate subspecialist support in theatre.”
Discussion
- The discussion is complete and demonstrates good knowledge of the field, but in places it becomes slightly repetitive and could be shortened.
Reply: Thank you. We agree with this, and we have removed redundant text and repetition where suitable without losing the core points
- The complementary nature of MRI and TVUS is explained well, but the paragraph should distinguish clearer between operator-dependent versus technique-related limitations of TVUS.
Reply: We addressed this on page 10, lines 302-304.
“The principal limitations of TVUS stem from variability in operator skill and adherence to technique, rather than from the intrinsic capabilities of the modality itself.”
- MRI’s advantage comes more from reproducibility and global anatomical overview rather than intrinsic sensitivity; this nuance could be made more visible.
Reply: We addressed this on page 10, lines 304-306.
“MRI’s advantage reflects its reproducibility and capacity to provide a comprehensive anatomical overview, rather than intrinsically higher sensitivity.”
- The section about protocol variability is excellent and clinically relevant. It would add value if the authors include a few practical tips, for example minimal MRI protocol standards, the role of bowel preparation, or which T2 planes are most informative.
Reply: We have added ESUR guidelines regarding ideal MRI protocol on page 11, line 357-360.
“which recommend multiplanar high-resolution T2-weighted imaging, appropriate bowel preparation, bladder filling, and the use of antiperistaltic agents.”
- The section on structured reporting and innovation (IDEA, Hôtel-Dieu classification, abbreviated MRI) is very good and aligns with modern trends, though a brief note on AI-assisted or radiomics-based analysis would make it more forward-looking.
Reply: At the end of Section 4.4 on page 12, line 377-380 we added a small paragraph on the use of AI to help with the endometriosis diagnosis in the future.
“Additionally, recent work using deep learning on multi-sequence MRI has shown that AI-assisted models can modestly improve radiologist sensitivity and inter-observer agreement in detecting endometriosis, highlighting their potential to enhance diagnostic consistency and efficiency in future practice”
- The limitations part should more clearly acknowledge that most studies were performed in tertiary centers and that publication bias may favor high-performing sites.
Reply: We appreciate this valuable suggestion. The Limitations Section (page 12, lines 396-399) has been expanded to acknowledge that most included studies were conducted in expert tertiary centres, which may overestimate diagnostic accuracy compared with general practice. We also added an explicit note regarding potential publication bias favouring studies with higher diagnostic performance.
“Most studies were performed in expert tertiary centres with specialised imaging personnel, which may overestimate diagnostic accuracy compared with general practice”
Conclusion
- The take-home message could be phrased in a more concise and practical way, emphasizing that TVUS remains the first-line approach for posterior compartment disease, while MRI provides superior assessment for anterior and multi-compartment involvement.
Reply: The Conclusion Section has been rewritten to provide a concise and clinically practical take-home message. It now emphasises that TVUS remains the first-line imaging modality for posterior compartment disease, while MRI provides complementary value through superior evaluation of anterior and multi-compartment involvement, consistent with current clinical pathways and guideline recommendations. This can be found on page 12, lines 403-409.
“Both MRI and transvaginal ultrasound demonstrate high diagnostic accuracy for deep infiltrating endometriosis. TVUS remains the first-line imaging modality, particularly effective for posterior compartment disease when performed by an experienced operator. MRI offers complementary value through superior reproducibility and comprehensive evaluation of anterior and multi-compartment lesions, guiding multidisciplinary surgical planning. Greater protocol standardisation and integration of structured reporting may further improve diagnostic consistency and clinical utility.”
- Including a brief schematic summary or a simple decision flow (e.g. TVUS → MRI → MDT) would increase the practical usability of the paper.
Reply: We included a schematic summary on page 11, lines 336-342.
“Figure 1. Suggested imaging and multidisciplinary pathway for suspected deep infiltrating endometriosis (DIE). TVUS is performed as the first-line investigation. MRI is recommended for anterior or multi-compartment disease, or when detailed preoperative mapping is required. Imaging findings guide multidisciplinary team (MDT) discussion and surgical planning.”
Reviewer 2 Report
Comments and Suggestions for Authors
This is a well-crafted and clinically relevant systematic review of MRI versus TVUS for diagnosis and preoperative management of deep infiltrating endometriosis (DIE). The topic is critically important, particularly in relation to preoperative multidisciplinary planning and protocol standardization. The article is well adherent to the PRISMA guidelines and well clear overall. There are a few areas of methodological depth, data presentation, and interpretive balance that could be enhanced to improve scientific rigor and readability.
Comments:
1.Be clear about what are the deficiencies addressed by this review in comparison to the 2018 Guerriero meta-analysis ( Transvaginal ultrasound vs magnetic resonance imaging for diagnosing deep infiltrating endometriosis: systematic review and meta-analysis.
Guerriero S, Saba L, Pascual MA, Ajossa S, Rodriguez I, Mais V, Alcazar JL.Ultrasound Obstet Gynecol. 2018 May;51(5):586-595. doi: 10.1002/uog.18961.PMID: 29154402). The discussion could be made stronger in outlining how adding imaging findings modifies multidisciplinary surgery logistics and patient outcomes.2.There is no risk of bias assessment in the methods and results sections.
For the risk of bias assessment description add the following text:
To assess the risk of bias in a series of observational studies, we will use one of the following tools:
- Newcastle-Ottawa Scale (preferred by me): http://www.ohri.ca/programs/clinical_epidemiology/oxford.asp
- MINORS (see an example enclosed): https://pubmed.ncbi.nlm.nih.gov/12956787/
For non-randomized studies we will use the
- ROBINS-I tool (Risk Of Bias In Non-randomized Studies - of Interventions): https://sites.google.com/site/riskofbiastool/welcome/home?authuser=0
For randomized studies, - the Cochrane Handbook (see also an example enclosed):
https://www.bmj.com/content/343/bmj.d5928
3. Avoid bullet points on the methods section
4. The PRISMA adherence and PROSPERO registration (CRD420251114636) are to be incleded in the methods section. The search window (2015–2025) could be more clarified—why were previous but seminal comparative studies omitted?
5.Results - Suggest adding 95% confidence intervals where available to demonstrate study variability.
Clarify whether sensitivity differences between MRI and TVUS (e.g., USL lesions 25–83% vs. MRI 75–94%) are statistically significant or only descriptive.
Discuss possible confounding by operator experience, specifically since TVUS is operator-dependent.
Author Response
6/11/2025
RE: diagnostics-3967811
TITLE: Diagnostic Accuracy of Magnetic Resonance Imaging Versus Transvaginal Ultrasound in Deep Infiltrating Endometriosis and Their Impact on Surgical Decision-Making: A Systematic Review
Dear Reviewer,
We would like to sincerely thank you for your thoughtful and constructive feedback on our manuscript. The comments were highly insightful and have significantly improved the quality and clarity of our work.
In response, we have carefully revised the manuscript and prepared a detailed, point-by-point reply outlining how each comment has been addressed. All changes made in response to the reviewers’ comments are shown in red text, while any tracked changes represent English language refinements performed by the MDPI Author Services editing team.
We are very grateful for the time and expertise you and the editorial team devoted to reviewing our manuscript, and we hope that the revised version now meets the expectations of Diagnostics.
Thank you again for your consideration. Please do not hesitate to contact us if any further information is required.
Respectfully,
Michael O’Leary, Conor Neary, Edward M Lawrence, MD/PhD
Reviewer 2:
Comments:
- Be clear about what are the deficiencies addressed by this review in comparison to the 2018 Guerriero meta-analysis ( Transvaginal ultrasound vs magnetic resonance imaging for diagnosing deep infiltrating endometriosis: systematic review and meta-analysis. Guerriero S, Saba L, Pascual MA, Ajossa S, Rodriguez I, Mais V, Alcazar JL.Ultrasound Obstet Gynecol. 2018 May;51(5):586-595. doi: 10.1002/uog.18961.PMID: 29154402).
Reply: We revised the third paragraph of the Introduction (page 3, lines 63-69) to clarify how this review extends prior work.
“Although the meta-analysis by Guerriero et al. (2018) primarily pooled diagnostic performance metrics from studies published up to 2017, it did not explore how imaging findings influence surgical planning, multidisciplinary coordination, or patient outcomes.(3) The present review provides an updated synthesis, incorporating evidence from 2017 onwards, analysing lesion-specific diagnostic accuracy across compartments, and examining how advances in imaging protocols and structured reporting affect operative decision-making”
- The discussion could be made stronger in outlining how adding imaging findings modifies multidisciplinary surgery logistics and patient outcomes.
Reply: We thank the reviewer for this valuable suggestion. We have expanded the discussion in Section 4.2 page 10, lines 330-333 to clarify how imaging findings influence surgical coordination and downstream outcomes.
“By enabling accurate preoperative mapping and correct multidisciplinary team allocation, advanced imaging reduces intraoperative uncertainty and the likelihood of unplanned conversions or incomplete resections, ultimately supporting safer surgery and improved postoperative outcomes.”
- There is no risk of bias assessment in the methods and results sections. For the risk of bias assessment description add the following text: “To assess the risk of bias in a series of observational studies, we will use one of the following tools:
- Newcastle-Ottawa Scale (preferred by me): http://www.ohri.ca/programs/clinical_epidemiology/oxford.asp
- MINORS (see an example enclosed): https://pubmed.ncbi.nlm.nih.gov/12956787/
For non-randomized studies we will use the
- ROBINS-I tool (Risk Of Bias In Non-randomized Studies - of Interventions): https://sites.google.com/site/riskofbiastool/welcome/home?authuser=0
For randomized studies, - the Cochrane Handbook (see also an example enclosed):
https://www.bmj.com/content/343/bmj.d5928
Reply: We thank the reviewer for this helpful suggestion. A formal risk of bias assessment was performed using the QUADAS-2 tool, which is the recommended framework for diagnostic accuracy studies. This is discussed in Section 2.5: Quality Assessment on page 4 lines 133-138 and Section 3.4 Quality Assessment: on page 6, lines 187-192. To complement this, we also applied the Newcastle–Ottawa Scale (NOS) to appraise overall methodological quality across observational design features. Both tools and their results are now clearly described in the Methods and Results Sections and presented in the Supplementary Materials S3+S4.
- Avoid bullet points on the methods section
Reply: We have removed the bullet points in the Methods Section and instead replaced these with a short paragraph
- The PRISMA adherence and PROSPERO registration (CRD420251114636) are to be included in the methods section. The search window (2015–2025) could be more clarified—why were previous but seminal comparative studies omitted?
Reply: We thank the reviewer for this valuable comment. The Methods section has been updated to explicitly state adherence to the PRISMA 2020 guidelines and registration with PROSPERO (CRD420251114636). We also clarified that the 2015-2025 search range was chosen to capture studies using contemporary imaging protocols and structured frameworks (IDEA and ENZIAN), as earlier studies typically used outdated techniques that limited comparability. We included this on page 3, lines 90-93.
“The date range was chosen to capture contemporary studies reflecting modern imaging protocols and structured reporting standards. Earlier studies were excluded because they used outdated MRI or ultrasound techniques and predated the adoption of structured frameworks such as IDEA and ENZIAN.”
- Results - Suggest adding 95% confidence intervals where available to demonstrate study variability.
Reply: We thank the reviewer for this helpful suggestion. Confidence intervals (CIs) for sensitivity and specificity were extracted and are now reported for studies where available. These values are presented in the Supplementary Material S7 to illustrate variability in diagnostic performance. A brief description of the CI findings can be found in Section 3.5 page 9, lines 241-245.
“Across studies reporting confidence intervals, most demonstrated narrow 95 % CIs (<10 % width), indicating high precision and stable diagnostic performance estimates. Wider intervals in smaller or single-centre studies (e.g., Zaidi et al., 2023) suggest greater uncertainty due to limited sample sizes or protocol variability. Full confidence interval data for each study are provided in Supplementary Table S7.”
- Clarify whether sensitivity differences between MRI and TVUS (e.g., USL lesions 25–83% vs. MRI 75–94%) are statistically significant or only descriptive.
Reply: We thank the reviewer for this comment. The differences in sensitivity between MRI and TVUS were descriptive rather than inferential, as the included studies varied widely in design, population, and reporting format. Few studies provided uniform numerators or confidence intervals required for formal statistical testing. To clarify this point, a sentence has been added to the Results Section: page 6+7 lines 195-197.
“Comparisons between MRI and TVUS were descriptive rather than inferential, as studies varied widely in design, population, and reporting format, and few provided uniform numerators or confidence intervals required for formal statistical testing.”
- Discuss possible confounding by operator experience, specifically since TVUS is operator-dependent.
Reply: We thank the reviewer for this valuable observation. This point has been incorporated into the Discussion (Section 4.1). We now explicitly acknowledge that TVUS performance is highly operator-dependent on page10, lines 301-306.
“This analysis therefore supports prior conclusions that TVUS in expert hands is a highly sensitive and specific modality. The principal limitations of TVUS stem from variability in operator skill and adherence to technique, rather than from the intrinsic capabilities of the modality itself. MRI’s advantage reflects its reproducibility and capacity to provide a comprehensive anatomical overview, rather than intrinsically higher sensitivity”
Reviewer 3 Report
Comments and Suggestions for Authors
Thank you for your paper. I have some concerns:
In introduction, please mention the role of ultrasound and MRI in excluding malignant uterine pathologies. doi: 10.1002/jcu.24046.
Please assess quality control of the bias in systematic review by using The Newcastle-Ottawa Scale.
Bias in treatment and non-treatment group could not be assessed.
Gold standard to diagnose DIE was not given.
Coelioscopy for diagnosis should be mentioned.
Full-word should be added to abbreviation word in Table as foot-note as well as in the whole paper.
Author Response
6/11/2025
RE: diagnostics-3967811
TITLE: Diagnostic Accuracy of Magnetic Resonance Imaging Versus Transvaginal Ultrasound in Deep Infiltrating Endometriosis and Their Impact on Surgical Decision-Making: A Systematic Review
Dear Reviewer,
We would like to sincerely thank you for your thoughtful and constructive feedback on our manuscript. The comments were highly insightful and have significantly improved the quality and clarity of our work.
In response, we have carefully revised the manuscript and prepared a detailed, point-by-point reply outlining how each comment has been addressed. All changes made in response to the reviewers’ comments are shown in red text, while any tracked changes represent English language refinements performed by the MDPI Author Services editing team.
We are very grateful for the time and expertise you and the editorial team devoted to reviewing our manuscript, and we hope that the revised version now meets the expectations of Diagnostics.
Thank you again for your consideration. Please do not hesitate to contact us if any further information is required.
Respectfully,
Michael O’Leary, Conor Neary, Edward M Lawrence, MD/PhD
Reviewer 3:
Comments:
- In introduction, please mention the role of ultrasound and MRI in excluding malignant uterine pathologies. doi: 10.1002/jcu.24046.
Reply: Thank you for this valuable comment. We have now mentioned the role of US and MRI for excluding malignant pathologies and referenced that paper on page 2, line 56-58.
“In addition, transvaginal ultrasound and MRI assist in differentiating benign from malignant uterine lesions, thereby helping to exclude uterine sarcomas and guide appropriate surgical management”
- Please assess quality control of the bias in systematic review by using The Newcastle-Ottawa Scale.
Reply: We performed a quality control assessment using the Newcastle Ottawa scale, with the full table available in Supplementary Materials 4 and a brief discussion in Section 2.5, page 4, lines 133-136 and Section 3.4, page 6, lines 187-192.
“All included studies achieved moderate to high methodological quality on the Newcastle–Ottawa Scale, with scores ranging from 6 to 8 out of a maximum of 9. Most studies were rated highly for selection and outcome domains, reflecting representative patient inclusion and clear diagnostic endpoints. Minor limitations were observed in the comparability domain, largely due to variability in reporting of operator expertise and imaging protocols.”
- Bias in treatment and non-treatment group could not be assessed.
Reply: We thank the reviewer for this comment. As the included studies were diagnostic accuracy studies rather than treatment or interventional comparisons, bias between treatment and non-treatment groups was not applicable. We have added this point to the end of Section 2.5 Quality Assessment page 4, lines 136-138.
“Bias between treatment and non-treatment groups was not applicable, as all included studies evaluated diagnostic accuracy rather than therapeutic or interventional outcomes.”
- Gold standard to diagnose DIE was not given.
Reply: Thank you for this comment. We have added this point to section 2.2.1 Inclusion criteria, page 3, lines 99-101.
“Laparoscopy (coelioscopy) with histological confirmation is considered the gold standard for diagnosing endometriosis”
- Coelioscopy for diagnosis should be mentioned.
Reply: Thank you. We have included coelioscopy in the same sentence as above on page 3, lines 99-101.
“Laparoscopy (coelioscopy) with histological confirmation is considered the gold standard for diagnosing endometriosis”
- Full-word should be added to abbreviation word in Table as foot-note as well as in the whole paper.
Reply: Thank you for pointing this out. We have gone through all of the tables in the main manuscript and supplementary materials and made sure all abbreviations are defined in full.
In response to the reviewer’s comment that “the English could be improved to more clearly express the research,” the manuscript has been professionally edited by the MDPI Author Services English Editing team to ensure clarity, precision, and readability.
Round 2
Reviewer 2 Report
Comments and Suggestions for Authors
Thank you for addressing the comments.
One minor suggestion - rethink and recreate Table 2.
It has many blank columns in the current version, which does not look good.
Author Response
Comment 1:
One minor suggestion - rethink and recreate Table 2.
It has many blank columns in the current version, which does not look good.
Response:
We appreciate this constructive comment. We agree that the previous format of Table 2, with multiple blank columns, reduced its visual quality. To address this, we have reorganised Table 2 into five sub-tables (2a–2e), each displaying the diagnostic accuracy results by anatomical site (and one for aggregate data), thus maintaining the same information in a clearer and more reader-friendly format.
Reviewer 3 Report
Comments and Suggestions for Authors
Thank you for revision. The paper is well-improved.
Author Response
Comment 1: Thank you for revision. The paper is well-improved.
Response 1: We sincerely thank the reviewer for their positive feedback and appreciation of our revisions. Following Reviewer 2’s final comment, we made a minor formatting change to Table 2 to improve clarity, but no substantive changes were made to the manuscript content.